# Novel *MAG* Variant Causes Cerebellar Ataxia with Oculomotor Apraxia: Molecular Basis and Expanded Clinical Phenotype

**DOI:** 10.3390/jcm9041212

**Published:** 2020-04-23

**Authors:** Mariana Santos, Joana Damásio, Célia Kun-Rodrigues, Clara Barbot, Jorge Sequeiros, José Brás, Isabel Alonso, Rita Guerreiro

**Affiliations:** 1UnIGENe, IBMC-Institute for Molecular and Cell Biology, i3S-Instituto de Investigação e Inovação em Saúde, Universidade do Porto, 4200-135 Porto, Portugal; mariana.graca@ibmc.up.pt (M.S.); joanadamasio80@gmail.com (J.D.); jorge.sequeiros@ibmc.up.pt (J.S.); 2CGPP, IBMC-Institute for Molecular and Cell Biology, i3S-Instituto de Investigação e Inovação em Saúde, Universidade do Porto, 4200-135 Porto, Portugal; clarabarbot@gmail.com; 3Neurology Department, Centro Hospitalar do Porto, 4099-001 Porto, Portugal; 4Center for Neurodegenerative Science, Van Andel Institute, Grand Rapids, Michigan, MI 49503, USA; Celia.Rodrigues@vai.org (C.K.-R.); Jose.Bras@vai.org (J.B.); 5Division of Psychiatry and Behavioral Medicine, Michigan State University College of Human Medicine, Grand Rapids, Michigan, MI 49503, USA

**Keywords:** cerebellar ataxia, myelin-associated glycoprotein, exome sequencing

## Abstract

Homozygous variants in *MAG*, encoding myelin-associated glycoprotein (MAG), have been associated with complicated forms of hereditary spastic paraplegia (HSP). MAG is a glycoprotein member of the immunoglobulin superfamily, expressed by myelination cells. In this study, we identified a novel homozygous missense variant in *MAG* (c.124T>C; p.Cys42Arg) in a Portuguese family with early-onset autosomal recessive cerebellar ataxia with neuropathy and oculomotor apraxia. We used homozygosity mapping and exome sequencing to identify the *MAG* variant, and cellular studies to confirm its detrimental effect. Our results showed that this variant reduces protein stability and impairs the post-translational processing (N-linked glycosylation) and subcellular localization of MAG, thereby associating a loss of protein function with the phenotype. Therefore, *MAG* variants should be considered in the diagnosis of hereditary cerebellar ataxia with oculomotor apraxia, in addition to spastic paraplegia.

## 1. Introduction

Hereditary cerebellar ataxias (HCAs) are a heterogeneous group of neurodegenerative disorders, characterized by motor incoordination and unsteady gait, often associated with speech and eye movement disturbances [1,2]. Autosomal recessive cerebellar ataxias are generally associated with peripheral sensorimotor neuropathy and non-neurological features. The most common of these is Friedreich ataxia, but recent advances in genetics, particularly the use of next-generating sequencing (NGS) technologies, have expanded the number of causative genes to more than 50 [3,4,5]. Additionally, HCAs share clinical features and disease mechanisms with hereditary spastic paraplegias (HSPs), with several ataxia-related genes found to cause HSPs and vice versa [6,7].

Homozygous variants in *MAG*, encoding the myelin-associated glycoprotein (MAG), have been associated with a complex neurological syndrome, including spastic paraplegia type 75 (SPG75), progressive neuropathy, ataxia and prominent sensorial dysfunction [8,9,10,11]. MAG, also known as Siglec-4a (sialic acid-binding immunoglobulin (Ig)-type lectin), is a transmembrane glycoprotein expressed by myelination cells, including oligodendrocytes and Schwann cells [12,13]. MAG is important for the maintenance of axon-glia attachment [14] and has a dual role in the inhibition of neurite outgrowth and axonal regeneration [15,16,17] and in neuroprotection from axonal injury [18,19].

In this study, we report a novel homozygous missense variant in *MAG* (c.124T>C; p.Cys42Arg) in a Portuguese family with early-onset autosomal recessive ataxia with neuropathy and oculomotor apraxia, identified by exome sequencing. Cell studies confirmed that the variant reduces protein stability, and impairs the post-translational processing and subcellular localization of MAG. Our findings thus suggest that this variant causes protein loss-of-function and associate *MAG* variants with ataxia with oculomotor apraxia (AOA).

## 2. Experimental Section

### 2.1. Genetic Analysis

A consanguineous Portuguese family affected by early-onset ataxia with oculomotor apraxia (AOA) was identified during a national systematic population-based survey (1994–2004) aiming to identify Portuguese families with HCA and HSP [20]. This survey included clinical history, family information, neurological evaluation and blood collection. Samples were collected after receipt of written informed consent from participants. In this study, we used only previously collected DNA samples. Variants in genes associated with Friedreich ataxia and the AOA phenotype (*APTX*, *SETX*, *PIK3R5* and *PNKP* genes) were excluded [20,21]; recently, the expansion of an intronic repeat in *RFC1* [22] was also excluded. 

Therefore, we performed whole-genome genotyping in two affected individuals using Illumina Infinium technology to identify the presence of large regions of homozygosity (>1 Mb). The samples were genotyped using the HumanOmniExpress-24v1-0_a BeadChip according to the manufacturer’s instructions, and data were visualized using the GenomeStudio Data Analysis Software (Illumina, San Diego, CA, USA). 

We also performed exome sequencing in the two affected individuals. Genomic DNA was prepared according to Illumina’s TruSeq Sample Preparation v.3, and exome capture was performed using Illumina’s TruSeq Exome Enrichment, according to the manufacturer’s instructions. Sequencing was performed on an Illumina HiSeq2500 with 100-bp paired-end reads. We performed sequence alignment and variant calling against the reference human genome (UCSC Human Genome Browser hg19) by using the Burrows-Wheeler Aligner [23] and the Genome Analysis Toolkit [24,25]. Prior to variant calling, PCR duplicates were removed with the Picard software. Given the apparent autosomal-recessive inheritance and consanguinity, we focused the analysis on homozygous variants located in loss of heterozygosity (LOH) regions. We filtered variants present in those regions using Exomiser v7.2.1 [26] with the following parameters: minor allele frequency (MAF) < 2%, autosomal recessive inheritance pattern, and human phenotype ontology HP:0001251 (term name: ataxia). Then, we excluded intronic, UTR, intergenic and synonymous variants and variants found in homozygosity in the Genome Aggregation Database (gnomAD; https://gnomad.broadinstitute.org). The functional predicted impact of variants was evaluated using the SIFT, PolyPhen-2, MutationTaster and CADD v1.5 software. 

We also used Sanger sequencing to confirm variants identified by exome sequencing and verified intrafamilial segregation. We performed PCR amplifications, using Ranger Mix (Bioline, London, UK) and purified products with Exo/SAP (GRiSP, Porto, Portugal), then performed Sanger sequencing using Big Dye Terminator Cycle Sequencing v1.1 (Applied Biosystems, Foster City, CA, USA) and an ABI 3130xl Genetic Analyzer (Applied Biosystems, Foster City, CA, USA). Sequencing analysis was carried out using the Seqscape v2.6 software (Applied Biosystems, Foster City, CA, USA). 

### 2.2. Antibodies

Primary antibodies: mouse monoclonal anti-EGFP antibody (MAB1765, Abnova, Taipei City, Taiwan), mouse monoclonal anti-GFP antibody (600-301-215, Rockland, Limerick, PA, USA), mouse monoclonal anti-GM130 antibody (610822, BD Biosciences, San Jose, CA, USA), and rabbit polyclonal anti-calnexin antibody (ADI-SPA-860, Enzo Life Sciences, Farmingdale, NY, USA). Secondary antibodies: horseradish peroxidase (HRP)-conjugated goat anti-mouse IgG (sc-2005, Santa Cruz Biotechnology, Heidelberg, Germany), HRP-conjugated goat anti-rabbit IgG (401393, Calbiochem, EDM Millipore, Darmstadt, Germany,), goat anti-mouse IgG Alexa Fluor® 568 conjugate (A-11004, ThermoFisher Scientific, Waltham, MA, USA), and goat anti-rabbit IgG Alexa Fluor® 568 conjugate (A-11011, ThermoFisher Scientific, Waltham, MA, USA). 

### 2.3. Expression Vectors

Human MAG cDNA was amplified from the pME18-MAG plasmid, kindly provided by Dr. Hisashi Arase [27], using the following primers: Forward 5’-GATCCTCGAGATGATATTCCTCACGGCACTG-3’ and reverse 5’- CGAGGAATTCTCTTGACCCGGATTTCAGC-3’. The purified PCR product was cloned into the pEGFP-N1 plasmid (Clontech, Mountain View, CA, USA) by restriction enzyme digestion (with XhoI and EcoRI, ThermoFisher Scientific, Waltham, MA, USA) and ligation with T4 DNA ligase (New England Biolabs, Ipswich, MA, USA). This plasmid was modified by site-directed mutagenesis, using the QuikChange II Kit (Agilent, Santa Clara, CA, USA) to produce disease-associated MAG plasmids. The following primers were used to introduce the C42R and S133R variants: Forward 5’-GCGTCTCCATCCCCCGCCGCTTTGACTTC-3’ and reverse 5’-GAAGTCAAAGCGGCGGGGGATGGAGACGC-3’ and forward 5’- CTTCTCAGAGCACAGGGTCCTGGATATCGTC-3’ and reverse 5’- GACGATATCCAGGACCCTGTGCTCTGAGAAG-3’, respectively. 

### 2.4. Cell Culture and Transfection

HEK293T cells (kindly provided by Elsa Logarinho, IBMC/i3S, Porto) were grown in DMEM high glucose GlutaMAX™ supplemented with 10% fetal bovine serum (FBS) and 1% antibiotic-antimycotic (Gibco, ThermoFisher Scientific, Waltham, MA, USA), at 37 °C, in a humidified 5% CO2 atmosphere. Cells were transiently transfected with each plasmid using jetPRIME (Polyplus-transfection, Illkirch, France) or Fugene® HD (Promega, Madison, WI, USA), according to the manufacturer’s protocol. 

In order to inhibit protein synthesis, cells were treated at 24 h post transfection with cycloheximide (100 µg/mL, Calbiochem, EDM Millipore, Darmstadt, Germany) and collected after 1, 3, 5, 7 or 24 h of incubation.

To inhibit/enhance different proteolytic pathways, cells were treated at 24 h post-transfection with the following drugs for 18 h: The proteasome inhibitor MG132 (5 μM, Calbiochem, EMD Millipore, Darmstadt, Germany), autophagy inducer rapamycin (200 mM, Calbiochem, EMD Millipore, Darmstadt, Germany), endoplasmic reticulum-associated protein degradation (ERAD) inhibitor EerI (10 µM, Calbiochem, EDM Millipore, Darmstadt, Germany) or control vehicle DMSO (Sigma-Aldrich, St. Louis, MO, USA), and the lysosomotropic agent NH4Cl (15 mM, Sigma-Aldrich, St. Louis, MO, USA) or control vehicle DMEM (Gibco, ThermoFisher Scientific, Waltham, MA, USA). 

### 2.5. Real-Time PCR

Total RNA was isolated from transfected cells, using NZYol (Nzytech, Lisbon, Portugal) as per the manufacturer’s recommendations, followed by the purification of the RNA aqueous phase, using the RNeasy mini kit (Qiagen, Hilden, Germany). RNA quantification was performed on NanoDrop 2000 (ThermoFisher Scientific, Waltham, MA, USA). cDNA was synthesized by the reverse transcription-PCR of 3 µg total RNA with oligo(dT), using the SuperScript III first-strand synthesis system (Invitrogen, Carlsbad, CA, USA), according to the manufacturer’s protocol. Templates were diluted 1000-fold and used with primers specific for human *MAG* (forward 5’-GATTATGATTTCAGCCTCC-3´’ and reverse 5´-CTATTGAAGTACCAGACAC-3’) and human β-actin (forward 5’-GCACTCTTCCAGCCTTCCTTC-3’ and reverse 5’-GTGATCTCCTTCTGCATCCTGTC-3’), along with the PowerUp™ SYBR™ Green Master Mix (Applied Biosystems, Foster City, CA, USA). All reactions were performed in triplicate and replicated three times, using the Applied Biosystems 7500 Fast Real-Time PCR (Applied Biosystems, Foster City, CA, USA). 

### 2.6. Deglycosylation Assay

Cells transfected with each plasmid for 48 h were collected in ice-cold 1× PBS with cOmplete^TM^ Protease Inhibitor Cocktail (Roche, Basel, Switzerland) and centrifuged at 500 *g* for 3 min. The supernatant was discarded and the pellet resuspended in glycoprotein denaturing buffer and heated at 100 °C for 10 min. Then, GlycoBuffer 3 with EndoH enzyme (New England Biolabs, Ipswich, MA, USA) was added, and the mixture was incubated at 37 °C for 3 h. The reaction was stopped by adding Laemmli buffer. Samples were further analyzed by Western blotting. 

### 2.7. Western Blot Analysis

Cells expressing target proteins were collected in RIPA buffer (150 mM NaCl, 1.0% IGEPAL® CA-630, 0.5% sodium deoxycholate, 0.1% SDS, 50 mM Tris, pH 8.0; Sigma-Aldrich, St. Louis, MO, USA) supplemented with cOmplete Protease Inhibitor Cocktail (Roche, Basel, Switzerland), except for deglycosylation and co-immunoprecipitation assays, and then sonicated. Total protein concentrations were measured with the Pierce BCA protein assay kit (ThermoFisher Scientific, Waltham, MA, USA), according to the manufacturer’s instructions. Samples (30 µg of total protein) were separated by SDS-PAGE and electrophoretically transferred onto PVDF membranes (Merck-Millipore, Darmstadt, Germany), using a wet electroblotting system (Bio-Rad, Hercules, CA, USA). These were stained with Ponceau S staining in order to assess equal gel loading and total protein quantification [28]. Then, membranes were blocked in 5% non-fat dry milk in TBS-T for 1 h, at room temperature (RT), and subsequently incubated with specific primary antibodies, as indicated (diluted in 3% non-fat dry milk in TBS-T), overnight at 4 °C. The membranes were washed with TBS-T and then incubated with HRP-conjugated secondary antibody for 1 h at RT. Following three washes with TBS-T, detection was achieved using WesternBrightTM Sirius (Advansta, San Jose, CA, USA), and chemiluminescence was detected with a ChemicDocTM XRS+ Imaging System (Bio-Rad, Hercules, CA, USA). Protein bands were quantified using the Image LabTM 5.2.1 Software (Bio-Rad, Hercules, CA, USA). Quantitative comparisons between the samples of each experiment were always performed on the same blot. When necessary, membranes were stripped by incubation in stripping solution (62.5 mM Tris-HCl, pH 6.7; 2% SDS; 100 mM beta-mercaptoethanol) at 50 °C for 30 min, with gentle agitation. 

### 2.8. Co-Immunoprecipitation

Cells transfected with each plasmid for 48 h were collected in lysis buffer (50 mM Tris-HCl, pH 8.0; 120 mM NaCl; 0.5% Triton X-100), supplemented with the Pierce Protease and Phosphatase Inhibitor tablets (ThermoFisher Scientific, Waltham, MA, USA), and sonicated. DynabeadsTM M-280 Sheep Anti-Mouse IgG (ThermoFisher Scientific, Waltham, MA, USA) were washed in 3% BSA/1× PBS. GFP antibody was coupled to Dynabeads (1 µg antibody/50 µL beads), by incubating with rotation overnight at 4 °C. Cell lysates were precleared with 5 μL Dynabeads for 1 h at 4 °C and then incubated with antibody-Dynabeads with rotation overnight at 4 °C. The immunoprecipitates were washed in 3% BSA/1× PBS and then in PBS, and further transferred to a clean tube. The proteins were eluted, by boiling in 1× Laemmli buffer at 90 °C for 10 min.

### 2.9. Immunofluorescence

Cells expressing target proteins were fixed using 4% paraformaldehyde/4% sucrose for 20 min and permeabilized with 0.3% triton X-100 for 20 min. After washing with PBS, cells were blocked in 3% BSA/PBS for 45 min and then further incubated with the primary antibodies overnight at 4 °C. Then, cells were incubated with the secondary antibodies for 1 h at RT, washed with PBS and stained with Hoechst 33342 (ThermoFisher Scientific, Waltham, MA, USA) for 5 min to label the nucleus. Preparations were mounted with ProLong^TM^ Gold Antifade Mountant (ThermoFisher Scientific, Waltham, MA, USA) and visualized using an epifluorescence Zeiss Axio Imager Z1 microscope equipped with an Axiocam MR3.0 camera and Axiovision 4.7 software (Carl Zeiss, Oberkochen, Germany).

### 2.10. Statistical Analysis

Analyses of data were performed using the IBM SPSS Statistics 25.0 software (IBM, Armonk, NY, USA). All quantitative data are expressed as mean ± standard deviation (SD) of at least three independent experiments. Statistical significance analysis was conducted using one-way ANOVA with Tukey’s post-hoc test; the level of statistical significance was set at *p* < 0.05. 

## 3. Results

### 3.1. Identification of a Novel MAG Variant in Ataxia Patients

The family studied here (Figure 1A) was identified during the national population-based survey (1994–2004), aiming to identify families with HCA and HSP [20]. In a first phase, variants in genes associated with the AOA phenotype and prevalent forms of recessive ataxia were excluded. Then, we performed whole-genome genotyping and exome sequencing (patients II.1 and II.3). We found 15 LOH regions shared by both patients (Appendix A). Then, we extracted the variants present in those regions and, after filtering, identified four homozygous variants in four genes (*MAG*, *LGI4*, *PLEKHG2* and *PLCE1*) (Appendix A). Only *MAG* had been associated with a clinical phenotype similar to that of the patients, as it is known to cause HSP that commonly overlaps with HCA [7]. Moreover, only the variant in *MAG* (NM_002361: c.124T>C; p.Cys42Arg; 4) was not reported in population databases, including gnomAD and the Nucleotide Polymorphism Database (dbSNP; https://www.ncbi.nlm.nih.gov/snp). We confirmed that both affected siblings carried the homozygous variant in *MAG* by Sanger sequencing (Figure 1B) and verified that the variant was present in heterozygosity in the non-affected mother (DNA from father not available), supporting intrafamilial segregation. In silico analysis with several pathogenicity prediction programs anticipated the amino acid substitution (p.Cys42Arg) to be deleterious (SIFT score = 0; Mutation Taster score = 1; Polyphen-2 score = 1; CADD_phred = 27.9). Moreover, the Cys42 residue of human MAG is conserved across different species (Figure 1C).

### 3.2. Clinical Features of the Patients Carrying the MAG Variant

The main clinical features and available medical history of the affected members from this family are summarized in Table 1. Briefly, all three patients presented ataxia as the first sign of disease at one year of age, had a very slow disease progression, despite severe motor handicap, and had longer survival. A delay in gait acquisition was present in all, with independent gait being achieved only after the age of three years. Early in the disease course, they presented upper limb dysmetria and mild intention tremor. Over the years, oculomotor apraxia emerged, and there was a progressive clinical deterioration, with severe ataxia, arreflexic tetraparesis, distal muscle wasting and extremely atrophic hands/feet. They all presented adult-onset optic atrophy. There was no evidence of cognitive impairment, though no formal neuropsychological assessment was performed. All three were wheelchair-bound between the third and fourth decades of life. In adulthood, the most prominent sign was a severe peripheral neuropathy. In addition to cerebellar atrophy on MRI, patient II:1 had spontaneous hyperintensity of the dentate nuclei, on T2-weighted images. EMG of the three siblings showed axonal neuropathy. Serum levels of alpha-fetoprotein, albumin and cholesterol were normal. The patients died after longstanding disabilities and long disease durations.

### 3.3. Cellular Characterization of the New MAG Variant

To understand the consequences of the p.Cys42Arg variant on the biochemical features of MAG, we generated constructs that direct the expression of wild-type MAG (MAG-WT) or MAG carrying the p.Cys42Arg variant (MAG-C42R) or the p.Ser133Arg variant (MAG-S133R), previously described in [8], in fusion with a C-terminal EGFP-tag. Constructs, including the control construct, were transfected into HEK293T cells. Immunoblotting analysis revealed that MAG-C42R had significantly reduced protein levels when compared with MAG-WT (Figure 2A, about a 67% decrease; *p* = 0.017). MAG-S133R had similar expression to MAG-C42R (about a 60% decrease compared with MAG-WT; *p* = 0.029). 

Moreover, while MAG-WT appeared as an apparent double band, indicating the presence of different MAG glycosylated forms, both MAG variants appeared as a single band, suggesting that the variants affected the post-translational processing of MAG. The incubation of cell lysates with EndoH showed that MAG-WT contained both high mannose N-linked glycans, which are removed by EndoH, and complex N-linked glycans (Figure 2B). MAG-C42R and MAG-S133R only contained high mannose N-linked glycans, as indicated by the downward shift after EndoH treatment (Figure 2B). The presence of high mannose glycans and absence of complex glycans indicate the endoplasmic reticulum (ER) retention of immature glycoproteins that have not trafficked to the Golgi [29].

The retention of MAG-S133R in the ER has been previously reported [8]. Therefore, we performed fluorescence microscopy to analyze the MAG-C42R distribution, and showed that it is very similar to that of MAG-S133R (Figure 3A). Cells labelled with an ER marker (calnexin) or a cis-Golgi marker (GM130) showed that MAG-C42R is located on the ER, while MAG-WT is predominantly found in the Golgi but is also scattered throughout the cell (Figure 3A). The immunoprecipitation of EGFP-tagged MAG proteins also demonstrated that only MAG-WT co-immunoprecipitated with GM130 (Figure 3B). On the other hand, calnexin was found in the immunoprecipitates of both MAG-WT and the MAG variants, but in higher levels in the latter (Figure 3B).

Given the low protein levels of MAG-C42R and to determine if the variant affects protein stability, we analyzed the half-life of MAG using a cyclohexamide chase assay. Our results showed that MAG-C42R had a higher degradation rate compared with MAG-WT (Figure 4). While the half-life of MAG-C42R was of about 3 h, MAG-WT levels were relatively stable (only decreased about 25%) after 24 h of cyclohexamide treatment (Figure 4A). Therefore, we treated cells with different drugs to identify the proteolytic pathway(s) responsible for MAG-C42R degradation. The treatment of cells with the proteasome inhibitor MG132 resulted in a significant increase in both MAG-WT and MAG-C42R protein levels (Figure 4B, *p* = 0.032 and *p* < 0.001, respectively). Even though the increase was substantially higher for MAG-C42R (an increase of about 250%, in contrast with the 40% increase for MAG-WT). The inhibition of autophagy nor ERAD—promoted by NH4Cl or EerI, respectively—nor the induction of autophagy by rapamycin significantly altered the levels of MAG-WT or MAG-C42R (Figure 4B). These results show that MAG-C42R is preferentially targeted to the proteasome for degradation, since only MG132 was capable of restoring protein levels. Of note, quantitative real-time PCR showed no significant reduction in MAG-C42R mRNA (Appendix A), confirming that decreased levels of MAG-C42R are not the result of a reduction in or instability of the mRNA.

## 4. Discussion

We identified a novel missense variant in *MAG* (c.124T>C; p.Cys42Arg) in homozygosity, in a consanguineous family presenting with cerebellar ataxia. The variant was identified using whole-genome genotyping and exome-sequencing, and functional studies in cells confirmed its detrimental effect on the MAG protein. Moreover, we showed that the biochemical and cellular characteristics of the MAG-C42R variant were very similar to those reported for a MAG-S133R variant [8].

Prominent cerebellar signs and oculomotor apraxia had prompted a clinical diagnosis of AOA, but genetic testing excluded pathogenic variants in known AOA genes [20,21]. Oculomotor apraxia is here described for the first time in association with *MAG* variants, broadening its clinical phenotype. In adulthood, the main clinical picture was of AOA with severe peripheral neuropathy and no pyramidal signs. As opposed to other AOA types [21,30], patients had no movement disorders, no suggestive biochemical anomalies, and all had optic neuropathy. When comparing with previously identified patients with *MAG* variants [8,9,10,11], the absence of pyramidal signs was a clear-cut difference. This could be due to the severe neuropathy obscuring pyramidal signs at clinical observation, though pyramidal tract degeneration could not be excluded. Additionally, none of the three patients showed apparent cognitive impairment. Nonetheless, there are some features overlapping with the previously reported *MAG*-related phenotype: delay of gait, cerebellar ataxia, peripheral neuropathy, optical nerve atrophy and cerebellar atrophy on MRI [8,9,10,11]. Moreover, all previous patients but one [9] presented their first symptoms very early in life (from their first days to around 3 years of age), and all had severe motor deficits, being wheelchair-bound or needing support for walking early in life. 

During our analysis, we also found homozygous variants in the genes *LGI4*, *PLEKHG2* and *PLCE1* (Appendix A), in addition to *MAG*; however, these genes have been associated with clinical features distinct to those of the patients of this study. Variants in *LGI4* cause arthrogryposis multiplex congenital; variants in *PLEKHG2* cause mental retardation, dystonia and microcephaly; and those in *PLEC1* have been shown to be involved in a renal disease (nephrotic syndrome). Variants in these genes are all described in population databases, except the one identified in *MAG*. Therefore, *MAG* was the main candidate gene in our study, as it is known to cause HSP that commonly overlaps with HCA [7].

The discovery of several genes causing both cerebellar and pyramidal signs has contributed to the understanding that HCA and HSP share many phenotypic but also pathophysiological features [6,7]. In addition, genes commonly associated with HSP, such as SPG7, were found to cause a predominant cerebellar ataxia phenotype in some patients lacking pyramidal signs [31,32]. NGS has been playing a major role in the discovery of new disease-causing genes and the broadening of the phenotypic spectrum of known ones, highlighting its power in the molecular diagnosis of these pathologies.

MAG is a component of myelin found in oligodendrocytes in the central nervous system and Schwann cells in the peripheral nervous system, important for the formation and maintenance of myelinated axons [12]. Thus, it is not surprising that some patients had white matter hyperintense lesions in T2 sequences, probably of a demyelinating nature, identified as soon as age 4 years [8,9,10]. Nonetheless, the two siblings described by Novarino and colleagues had normal brain MRI, which could be due to the younger age at the time of the exam [11]. None of our patients had white matter lesions, but one had a bilaterally increased signal at the dentate nucleus, in T2 images (not shown). Due to its high iron content, the dentate nuclei usually appears hypointense in T2; hyperintensity is described in a variety of acquired and genetic disorders, such as multiple sclerosis, Leigh syndrome or glutaric aciduria type 1 [33]. Over the past years, it has also been described in degenerative diseases, such as SCA48 and SPG7 [34,35].

Interestingly, the process of myelination is not affected in MAG-deficient mice, which show subtle morphological abnormalities in myelin. It seems that MAG is not critical for myelin formation, but is needed for the maintenance of myelinated axons [36,37]. 

Structurally, MAG is a type I transmembrane protein, consisting of an N-terminal extracellular region with five disulfide-bonded Ig-like domains, a single transmembrane domain and a short cytoplasmic region (Figure 1C). The first Ig-like domain (Ig1), where the MAG-C42R variant is located, has a V-type Ig fold like other Siglec family members, is conserved across sialoproteins, and is critical for sialic acid binding and interaction with axons. Moreover, Cys42 and Cys100 are predicted to form a disulphide bridge [13,38]; thereby, the replacement of Cys42 will abolish the cysteine bridge (Appendix A), probably altering the structure of the protein and thus disturbing its stability [39].

Accordingly, our results demonstrated that the MAG-C42R variant affects the protein’s stability, post-translational modification (glycosylation) and subcellular distribution. Cyclohexamide assays demonstrated that MAG-C42R undergoes rapid degradation, when compared to MAG-WT, which was relatively stable (Figure 4A). Moreover, we showed that this premature degradation that causes lower levels of MAG-C42R protein (Figure 2A) is mainly due to proteasome-dependent degradation (Figure 4B). However, the mutated protein was still expressed at moderate levels when overexpressed in cells, raising the question if the endogenous protein could be fully degraded by the proteasome in patients’ cells. Indeed, other authors reported similar results for overexpressed MAG-S133R variant [8]. Moreover, they also demonstrated that MAG is absent in the myelin sheath from sural nerve biopsies. Unfortunately, we did not have access to patients´ cells or tissues to confirm this. 

MAG is a heavily glycosylated protein and contains eight N-linked glycosylation sites scattered along the extracellular region [40]. The initial modification of glycans is accomplished in the ER; properly folded glycoproteins then move to the Golgi to generate the complex glycans found in mature glycoproteins. It is thought that N-linked glycans provide highly hydrophilic groups that are important for proper folding and the modulation of protein conformation [29,41]. Our study demonstrated that while MAG-WT contained both high mannose N-linked glycans and complex N-linked glycans, MAG-C42R contained only high mannose N-linked glycans that are sensitive to EndoH treatment (Figure 2B). The absence of complex glycans suggests that MAG-C42R is retained in the ER and does not traffic to the Golgi for further modification. Thus, our results indicate that MAG-C42R stays in the ER for further degradation, as it is not properly folded. In line with this, we showed that both MAG-C42R and MAG-S133R are present in the ER and interact with calnexin (Figure 3), which participates in the folding of glycoproteins and prevents the export of improperly folded proteins from the ER [29,41]. Thus, it is not surprising that calnexin also binds to MAG-WT, though apparently with lower affinity than to MAG-C42R or MAG-S133R. Moreover, we also showed that MAG-WT traffics to the Golgi complex, where it associates with GM130 (Figure 3), a cis-Golgi matrix protein. Additionally, we did not find evidence of MAG-C42R degradation by the ERAD, as the treatment of cells with the EerI drug (which associates with VCP to inhibit ERAD) did not significantly change the levels of the protein.

In summary, we identified and characterized a novel variant in *MAG* and expanded the clinical phenotype associated with its disease-causing variants. Therefore, *MAG* should be considered in the diagnosis of recessive ataxia with oculomotor apraxia. The low expression and premature degradation of the mutant protein supports a loss-of-function phenotype. 

## Figures and Tables

**Figure 1 jcm-09-01212-f001:**
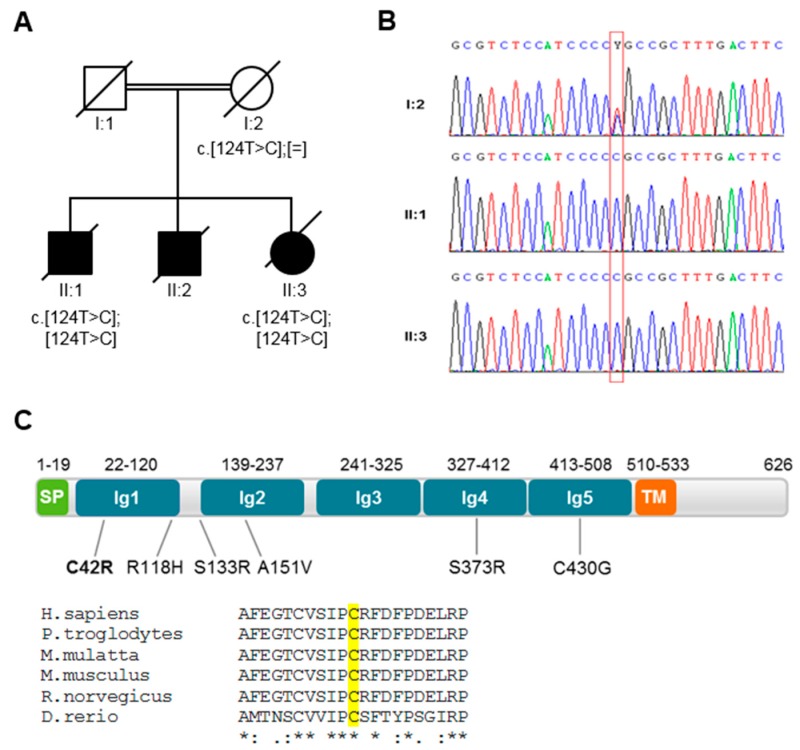
A family pedigree and schematic representation of the myelin-associated glycoprotein (MAG) protein. (**A**) A pedigree with segregation analysis of the new *MAG* variant (c.124T>C; p.Cys42Arg). Black symbols represent individuals affected by cerebellar ataxia. (**B**) An electropherogram with the position of the c.124C>T variant boxed. (**C**) A schematic representation of the MAG protein (NP_002352.1) with the location of the variant found in this study (in bold) and other variants previously associated with different clinical phenotypes. A sequence alignment of the residues surrounding Cys42 of human MAG against other species is also shown and was performed using the Clustal Omega program.

**Figure 2 jcm-09-01212-f002:**
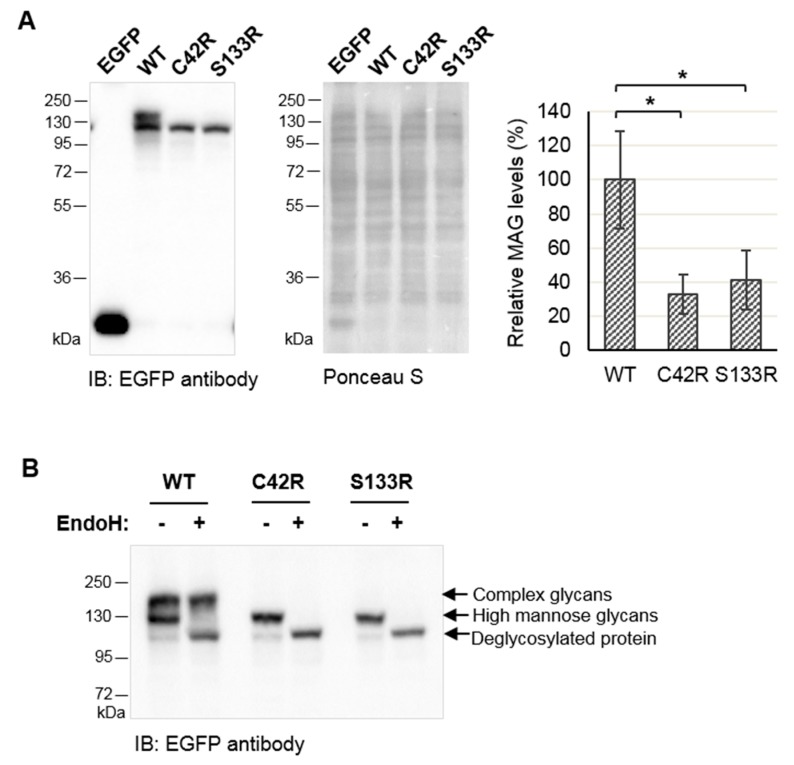
The MAG-C42R variant results in reduced protein expression and impairs glycosylation. (**A**) Analysis of protein expression of EGFP-tagged MAG clones in HEK293T cells by immunoblotting with anti-EGFP antibody. Ponceau S staining was used as the loading control. Quantification data (graph) are presented as the mean ± SD of three independent experiments; * *p* < 0.05, compared with MAG-WT (one-way ANOVA/Tukey). (**B**) Cell lysates were incubated with EndoH, which cleaves high mannose N-linked glycans but not complex N-linked glycans, and subjected to immunoblotting with anti-EGFP antibody. The original blot is presented in Appendix A.

**Figure 3 jcm-09-01212-f003:**
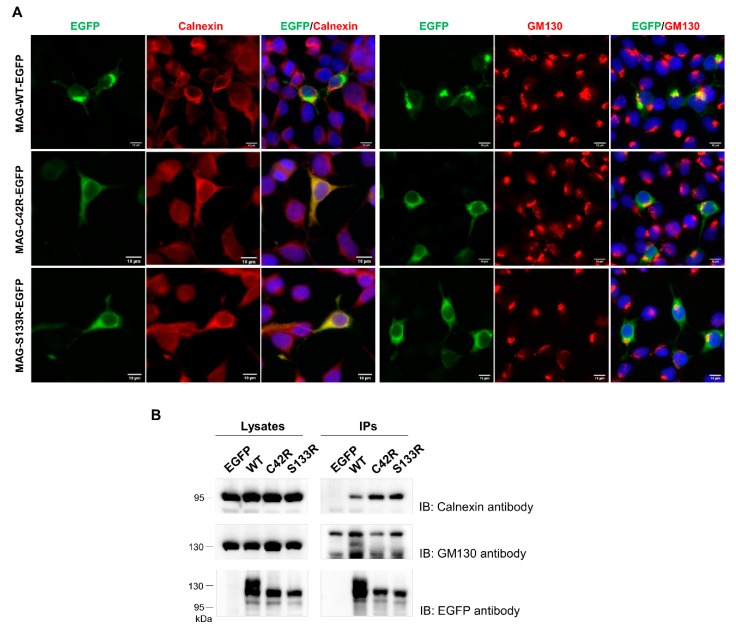
MAG-C42R has an altered subcellular localization. (**A**) The localization of EGFP-tagged MAG clones in HEK293T cells. EGFP was directly visualized. The presence of MAG in the endoplasmic reticulum and Golgi complex was analyzed by probing cells with anti-calnexin and anti-GM130 antibodies, respectively, both detected with Alexa Fluor 568-conjugated secondary antibodies (red). DNA was stained with Hoechst 33342 (blue). Photographs were acquired using a Zeiss Axio Imager Z1 microscope. Bars: 10 μm. (**B**) The co-immunoprecipitation of EGFP-tagged MAG clones with endogenous calnexin and GM130 in HEK293T cells. Cell lysates were immunoprecipitated with anti-GFP antibody. The immunoprecipitates were subjected to immunoblotting with anti-GM130 antibody, then stripped and reprobed with anti-calnexin antibody, followed by stripping and reprobing with anti-EGFP antibody. The same methodology was applied to whole cell lysates. Calnexin and GM130 immunoblotting figures have different exposure times (original blots are presented in Appendix A).

**Figure 4 jcm-09-01212-f004:**
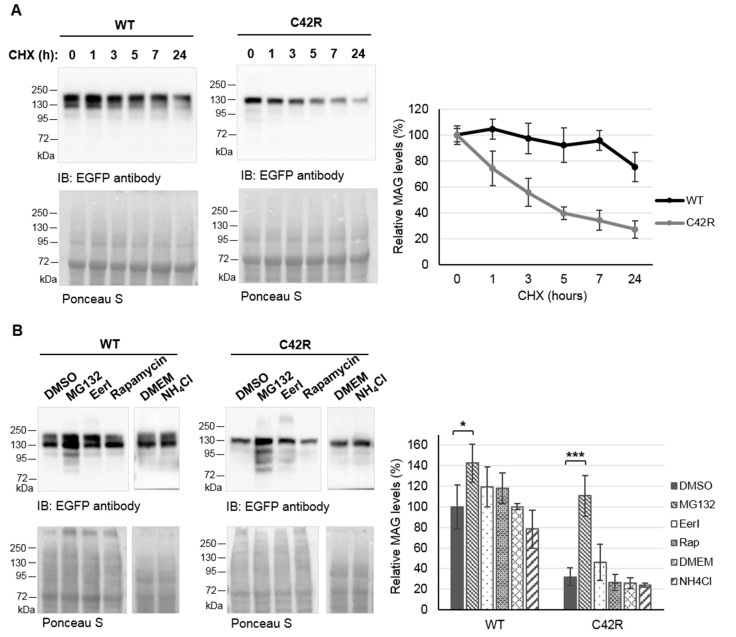
MAG-C42R impairs protein stability and is degraded by the proteasome. Cells expressing EGFP-tagged MAG clones were incubated with cycloheximide (CHX) for the indicated periods of time (**A**) or with different drugs for 18 h (**B**). Membranes were stained with Ponceau S for total protein quantification (whole blots are presented in Appendix A) and then cut and probed with anti-EGFP antibody. Quantification data are presented as the mean ± SEM (**A**) or mean ± SD (**B**) of three independent experiments; * *p* < 0.05, *** *p* < 0.001 (one-way ANOVA/Tukey).

**Table 1 jcm-09-01212-t001:** The clinical, biochemical and imaging features of the patients carrying the *MAG* variant.

	II:1	II:2	II:3
Gender	Male	Male	Female
Age of onset (years)	1	1	1
Last examination (years)	59	56	54
Age of death (years)	63	56	7th decade
First sign	Ataxia	Ataxia	Ataxia
Prominent sign	Neuropathy	Neuropathy	Neuropathy
OA	+	++	++
Wheelchair use (years)	20	21	30
Dystonia	-	-	-
Cognitive impairment	-	-	-
Motor deficit	+++	+++	+++
Spasticity	-	-	-
Pyramidal signs	-	-	-
Limb edema	-	-	-
Scoliosis	-	-	-
*Pes cavus*	+	+	+
Optic atrophy	+	+	+
Obesity	-	-	-
MRI findings	Cerebellar atrophy. Dentate nuclei hyperintensity on T2 sequences	Cerebellar atrophy	Cerebellar atrophy
EMG findings	Axonal neuropathy	Axonal neuropathy	Axonal neuropathy
VEP	Optical nerve damage	Optical nerve damage	Optical nerve damage
Alpha-fetoprotein levels	N	N	N
Albumin levels	N	N	N
Cholesterol levels	N	N	N

+, present; ++, moderate; +++, severe; -, absent; OA, oculomotor apraxia; N, normal levels; EMG, electromyography; VEP, visual evoked potentials.

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
