# Peer review of "Novel MAG Variant Causes Cerebellar Ataxia with Oculomotor Apraxia: Molecular Basis and Expanded Clinical Phenotype"

_jcm, 2020, doi:10.3390/jcm9041212_

Round 1

Reviewer 1 Report

Dear Editor,

In this study Santos et al identified a novel homozygous missense variant in MAG gene in 2 siblings with cerebellar ataxia, born to consanguineous parents. They provide evidence of the pathogenic role of the variant. In particular, they show that, similarly to the already known mutation, the variant reduces the levels of protein stability leading to increased proteasome degradation (normal mRNA levels, reduced protein levels) and impairs post-translational processing and subcellular localization. This finding broadens the spectrum of clinical manifestations associated with MAG alterations. The paper is well written and deserves consideration for the publication in its current form.

No mention to informed consent has been given.

The only important point is to specify the reason of not testing the third sibling.

Author Response

We thank the reviewers for their constructive comments. We have taken the comments into account to improve and clarify the manuscript. Please find below a detailed point-by-point response to the comments.

No mention to informed consent has been given.

Response: We modified and inserted a new sentence (lines 59-60) referring to the informed consent, that was not clearly explicit in the previous form.

“This survey included clinical history, family information, neurological evaluation and blood collection. Samples were collected after receipt of written informed consent from participants. In this study, we used only previously collected DNA samples.”

The only important point is to specify the reason of not testing the third sibling.

Response: During the population-based survey, the clinicians performed the study of this family in two phases. On a first visit, they collected the clinical history and family information, and performed the neurological evaluation of all family members that consented the study. On a second visit, they collected blood samples for genomic DNA extraction. Unfortunately, during this process, the clinical condition of the third sibling had deteriorated and he died. Thus, it was not possible to collect blood for testing.

Reviewer 2 Report

In this study, Santos and colleagues identify a novel homozygous missense variant in the MAG gene (c.124T>C; p.Cys42Arg) in a Portuguese family with early-onset, autosomal recessive cerebellar ataxia with neuropathy and oculomotor apraxia.  They used homozygosity mapping and exome sequencing to identify the MAG missense variant, and they performed cellular and biochemical studies to confirm its pathogenic effect. 

Overall, I found this to be a well-written, compelling and thorough report, wiht well-described methods.  

My comments are all minor:

  1. Line 42, HSP75 shoudl be listed as SPG75.
  2. Line 133, minor typo:  'cOmplete' shoudl be 'Complete'
  3. Lines 137 and 165, 'laemmli' should be 'Laemmli'
  4. Line 270/table,  could the authors provide the specific ages of death
  5. For Figure 3A the colocalizations can be difficult to see in the merged image. Could the authors add in panels with just the red channels to ease comparisons?

Author Response

We thank the reviewers for their constructive comments. We have taken the comments into account to improve and clarify the manuscript. Please find below a detailed point-by-point response to the comments.

1. Line 42, HSP75 shoudl be listed as SPG75.

Response: We modified HSP75 to SPG75 as indicated.

2. Line 133, minor typo:  'cOmplete' shoudl be 'Complete'

Response: We added the trademark ™ (“cOmplete™ Protease Inhibitor Cocktail). This is the commercial name of the product.

3. Lines 137 and 165, 'laemmli' should be 'Laemmli'

Response: We modified laemmli to Laemmli as indicated.

4. Line 270/table,  could the authors provide the specific ages of death

Response: We added the age of death in table 1 as suggested. The patient II.3 went to a care facility after the last examination by the clinicians; thus, we only have the information that she died during the seventh decade of life (ages 60-69).

5. For Figure 3A the colocalizations can be difficult to see in the merged image. Could the authors add in panels with just the red channels to ease comparisons?

Response: As suggested, we added the red channels to figure 3A. We hope that it improves the evaluation of the protein’s localization.